# Explainabilty Comparison between Random Forests and Neural Networks—Case Study of Amino Acid Volume Prediction

**Roberta De Fazio** [1,†] , **Rosy Di Giovannantonio** [1,†] , **Emanuele Bellini** [2,†] **and Stefano Marrone** [1,*,†]

1 Dipartimento di Matematica e Fisica, Università degli Studi della Campania "Luigi Vanvitelli", Viale Lincoln, 5-81100 Caserta, Italy

2 Dipartimento di Studi Umanistici, Università degli Studi Roma Tre, Via Ostiense, 234-00146 Roma, Italy

* Correspondence: stefano.marrone@unicampania.it; Tel.: +39-0823-27-5101

† All the authors contributed equally to this work.

**Abstract:** As explainability seems to be the driver for a wiser adoption of Artificial Intelligence in healthcare and in critical applications, in general, a comprehensive study of this field is far from being completed. On one hand, a final definition and theoretical measurements of explainability have not been assessed, yet, on the other hand, some tools and frameworks for the practical evaluation of this feature are now present. This paper aims to present a concrete experience in using some of these explainability-related techniques in the problem of predicting the size of amino acids in real-world protein structures. In particular, the feature importance calculation embedded in Random Forest (RF) training is compared with the results of the Eli-5 tool applied to the Neural Network (NN) model. Both the predictors are trained on the same dataset, which is extracted from Protein Data Bank (PDB), considering 446 myoglobins structures and process it with several tools to implement a geometrical model and perform analyses on it. The comparison between the two models draws different conclusions about the residues' geometry and their biological properties.

**Keywords:** random forest; multi-layer perceptron; explainable AI; protein data bank; neural network; machine learning





## 1. Introduction

Modern society and industry are demanding more and more smart applications, based on the paradigm of Artificial Intelligence (AI) [1]; the advantages span from a higher competitiveness of companies to the possibility of building more sustainable smart cities [2]. In particular, the Machine Learning (ML) paradigm is promising for solving problems whose solution is too difficult to be expressed in traditional algorithms: the availability of a sufficient amount of data can—in theory—allow solving complex problems with "zero-knowledge". As hunger for smart applications increases, critical domains are starting to be affected by this new software engineering paradigm: autonomous driving [3], Clinical Decision Support System (CDSS) [4] and financing-related applications [5] are just a few of these critical domains.

Notwithstanding the push of the market and the society, the applications of such a paradigm in critical domains is far from being simple; this is because the operation of safety-critical, business-critical and privacy-critical applications is based on rigorous and repeatable Verification & Validation (V&V) processes [6,7]. Such processes are mainly based on modelling, static and dynamic analyses of software, which require having a clear view of the actual behaviour of the code. In contrast, in ML-based code the application emerges from the weights between model parts. Explainability has risen in these years as a must for critical applications of AI; since NNs appear like black box phenomena, the behaviour of algorithms needs to be explained and rebuilt to make sense of the results [8]. As defined in [9], given an audience, an explainable Artificial Intelligence is one that produces details or reasons that make its functioning clear or easy to understand .

The objective of the paper is to report an experience in comparing two of the most widespread ML models under the lens of explainability. In particular, the models of RFs and NNs have been chosen and tested on a challenging case study: the prediction of volume of protein residues. The experience described, starts with the extraction of a subset from the PDB, including the group of myoglobin proteins—that will be preprocessed according to a method already present in the literature [10–12]—and then fed to the two different ML models, according to a set of geometric and non-geometric features. The performances of trained models are then analyzed with existing tools to explore which are—according to the different explainability tools—the most impactful and important features for the prediction task.

> It is important to underline that this paper focuses on the ML approaches and has the primary and sole purpose of comparing the presented approaches on the base of a replicable case study. The case study is taken as a driver to demonstrate the results of such a comparison: the authors are aware that the protein volume prediction—as taken as a problem itself—would need a more complex approach and the application of sophisticated methods and techniques that are not in the scope of the present paper.

The most valuable contribution of this paper is constituted by dealing with a challenging real case study. Many scientific papers report theoretical evaluations of the different ML-based explainability techniques [13,14]. The approach followed in this paper is different, preferring to focus on the reporting of a practical but repeatable experience in comparing off-the-shelf methods and technologies, rather than defining ad hoc solutions. Such a comparison is then conducted on a real-world case study.

This paper is structured as follows: Section 2 gives some background information about PDB, the used ML models and the related mechanisms to provide explainability facilities. Section 3 describes the methodology followed. Section 4 focuses on the preprocessing phase—which is always a foundational and crucial phase in every ML-based approach—highlighting the critical steps. Section 5 reports the results of the models training, while Section 6 compares the results under the different considered aspects. Section 7 gives a brief review of the related works, while Section 8 ends the paper and lays out future research lines.

## 2. Background

This section recalls some background concepts, reported for clarity. Section 2.1 reports the main concepts of the PDB and related manipulating software libraries, Section 2.2 recalls the base concepts of the used ML models and techniques, Section 2.3 gives some elements of the Eli-5 software library.

### 2.1. Protein Data Bank

PDB is a key resource in structural biology; it was created in 1971 and, since that date, it has been extensively used in international research projects [15]. It contains information about the exact location of all the atoms in more than 195,565 protein structures identified by a four-letter alphanumeric code. The structures are determined using different methods—e.g., electron microscopy, X-ray diffraction—and coded using a file format, considering information as name and function of the protein, the organism to which it belongs, crystallographic properties, quality of the structure, bibliographic references of the study, classification.

One of the most used notations concerning which the proteins contained in PDB can benefit is the textual notation. Each line protein is called a record; the different types of records are arranged in a specific order to describe a structure. Listing 1 reports an excerpt of the PDB structure of the Ferric Horse Heart Myoglobin; H64V/V67R Mutant (PDB code: 3HEN) [16].

**Listing 1.** PDB code: 3HEN protein file excerpt.

```
HEADER    OXYGEN TRANSPORT                       09-MAY-09   3HEN
TITLE     FERRIC HORSE HEART MYOGLOBIN; H64V/V67R MUTANT
COMPND    MOL_ID: 1;
COMPND   2 MOLECULE: MYOGLOBIN;
SOURCE   2 ORGANISM_SCIENTIFIC: EQUUS CABALLUS;
SOURCE   3 ORGANISM_COMMON: DOMESTIC HORSE, EQUINE;
SEQRES    1 A  153   GLY LEU SER ASP GLY GLU TRP GLN GLN VAL LEU ASN VAL
SEQRES    2 A  153   TRP GLY LYS VAL GLU ALA ASP ILE ALA GLY HIS GLY GLN
SEQRES    3 A  153   GLU VAL LEU ILE ARG LEU PHE THR GLY HIS PRO GLU THR
SEQRES    4 A  153   LEU GLU LYS PHE ASP LYS PHE LYS HIS LEU LYS THR GLU
SEQRES    5 A  153   ALA GLU MET LYS ALA SER GLU ASP LEU LYS LYS VAL GLY
SEQRES    6 A  153   THR ARG VAL LEU THR ALA LEU GLY GLY ILE LEU LYS LYS
SEQRES    7 A  153   LYS GLY HIS HIS GLU ALA GLU LEU LYS PRO LEU ALA GLN
SEQRES    8 A  153   SER HIS ALA THR LYS HIS LYS ILE PRO ILE LYS TYR LEU
SEQRES    9 A  153   GLU PHE ILE SER ASP ALA ILE ILE HIS VAL LEU HIS SER
SEQRES   10 A  153   LYS HIS PRO GLY ASP PHE GLY ALA ASP ALA GLN GLY ALA
SEQRES   11 A  153   MET THR LYS ALA LEU GLU LEU PHE ARG ASN ASP ILE ALA
SEQRES   12 A  153   ALA LYS TYR LYS GLU LEU GLY PHE GLN GLY
ATOM      1  N   GLY A   1      -1.476  41.015 -11.482  1.00 40.53           N
ATOM      2  CA  GLY A   1      -2.113  40.213 -12.574  1.00 40.50           C
ATOM      3  C   GLY A   1      -1.163  40.052 -13.757  1.00 38.97           C
ATOM      4  O   GLY A   1      -0.026  40.555 -13.734  1.00 40.91˜O
```

As an example, each ATOM record represents the location, represented by x, y, z orthogonal coordinates, occupancy and temperature factor of each atom of the protein. HELIX and SHEET indicate the location and type of helices or sheet in the secondary structure. SEQRES, instead, contains the primary sequence of amino acids that belong to the protein.

### 2.2. Random Forest and MLP

The RF technique, extension of the construction approach of decision trees, belongs to the class of Average Ensemble methods [17]. The idea is based on the construction of several different independent estimators and, for all of them, to calculate an average of all predictions. The combined estimator indeed is often better than any single estimator, since it will have a reduced variance.

To overcome the limits of the perceptron, a more complex structure was introduced. One or more intermediate levels were added within Neural Networks, creating a class called Multilayer Perceptron Neural Network (MLP) [18]. The new model has three layers: input layer, output layer and hidden layer; in these networks, the signals travel from the input layer to the output layer and therefore they are also called multi-layer feed-forward networks. Each neuron, in a generic layer, is connected with all those of the next layer, so the propagation of the signal occurs forward in an acyclic way and without transverse connections.

### 2.3. The Eli-5 Tool

The most controversial aspect of using MLP concerns the problem of the network behaviour interpretability. Neural Networks have always been considered like a sort of black box: they use an advanced technique in pattern recognition based on a strong algorithm of optimization, but they are not based on a structured model, so their results have to be explained. Eli-5 is a Python library that allows one to rebuilt the network behaviour using the Mean Decrease Accuracy algorithm [19–21]. This algorithm is based on the calculation and comparison of several scores achieved by the network in data prediction during some training, each of which is performed without a particular feature of the dataset. At the end of the algorithm, every feature will have its score of importance in the prediction of the target variable; that is, the higher it is, the lower the score of prediction performed by the network without that feature in the dataset. The only con of this algorithm is the required computational cost because it needs retraining of the network for each feature of the dataset.

## 3. Description of the Methodology

Figure 1 sketches the schema of the methodology followed.

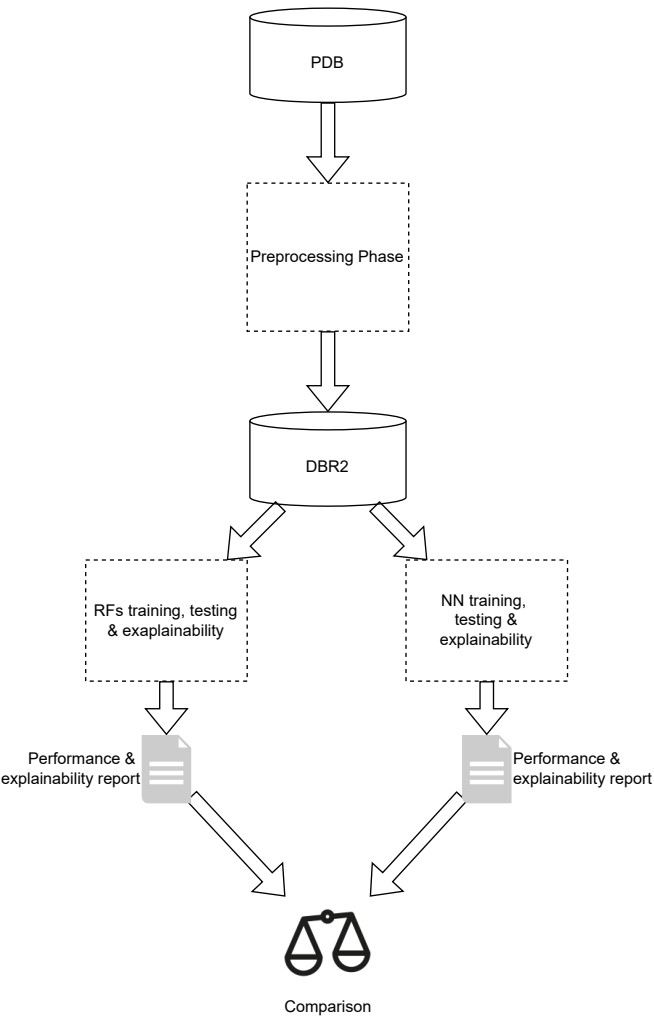

**Figure 1.** The schema of the methodology.

The main goal of this paper is comparing different analysis techniques in a particular case of study, in the field of bioinformatics. To pursue such goal, it is necessary to understand the background, explained in Section 2. The first phase concerns the study of the whole PDB, performing a qualitative analysis to define the subset of such a database that is the subject of the study.

The second phase sees a set of pre-processing tasks oriented to engineer and organize data from the PDB to extract the most meaningful features for the proposed analysis. The most meaningful tasks of such a phase are the definition and the design of a relational data base (DB) schema, the extraction and population processes oriented to import data from PDB files into such a DB, the transformation of such data to extract geometrical features. The tools used to face these challenges are the DBMS PostgreSQL and the Python libraries BioPython [22] and DSSP [23]. The technical details and the results of such a phase are reported in Section 4. Such results are contained in the Rosy and Roberta Database (DBR$^2$) database (both schema and instance).

Then the third phase involves two parallel activities that use the feature set defined during the first phase, to train two different amino acid volume predictors: the first using the RF model and the second using the MLP. Both the models are trained and tested starting from data contained in DBR$^2$ and all the details are explained in Section 5. On the trained

model, explainability tools are used to obtain the most important features. The results of these activities are contained in two performance and explainability reports.

The final step of the approach consists in a comparison between the two methods, discussing the differences and the common points. Section 6 is devoted to this step.

## 4. From PDB to Geometrical Data

### 4.1. Data Preparation Process

In a general overview of the work done, the collection of the initial data and the careful study of these constitute the most important starting point. In this case study, information comes from the same source, the PDB. It was decided to focus on a particular biological family of proteins: the myoglobins. The initial design phase produces a conceptual schema, implemented in the form of a physical schema. The first population is based on data extraction from PDB files, using the Python library called BioPython. Then, an appropriate geometric model is constructed to describe the spatial properties of the protein primary sequence. Using this model, data are manipulated to proceed with a second population. Figure 2 illustrates these steps.

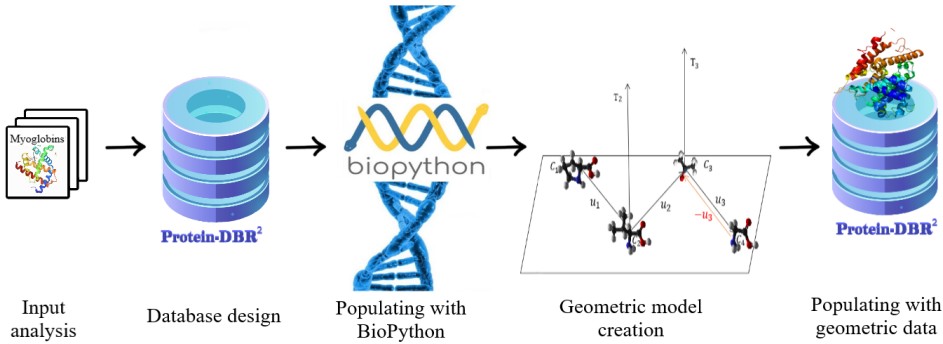

Input analysis | Database design | Populating with BioPython | Geometric model creation | Populating with geometric data

**Figure 2.** Process to realize the database management system.

### 4.2. Protein Geometrical Modelling

This study is based on a mathematical model designed for the description and prediction of the spatial structure of a protein. New mathematical tools were introduced to adequately represent one-dimensional and three-dimensional structures of proteins, so these can be regarded as mathematical objects. For further details or explanations, please refer to the publication [10,12].

As is well known, the notion of curvature is characteristic of continuous curves; in this particular case, it was necessary to introduce a discretized version of this parameter for the interpretation of a protein in its natural configuration as a folded chain. Thanks to this, it was possible to re-define torsion as a vectorial product between two curvatures as follows:

**Definition 1.** $\forall h \in (1, \ldots, n-1)$ *consider the box* $B_h$ *of the h-th amino acid* $s_p^h$ *of a given protein P. The geometric centre vector:*

$$C_h \equiv \left( \frac{x_i^* + x_{*i}}{2}, \frac{y_i^* + y_{*i}}{2}, \frac{z_i^* + z_{*i}}{2} \right) \tag{1}$$

*is related to the α-Carbon position in the residue.*

The vector that links geometric centres of two following amino acids $s_P^h$, $s_P^{h+1}$ is given by:

$$\mathbf{u}_h \equiv C_{h+1} - C_h \equiv \tag{2}$$

$$\equiv \left( \frac{x_{h+1}^* + x_{*h+1}}{2} - \frac{x_h^* + x_{*h}}{2}, \frac{y_{h+1}^* + y_{*h+1}}{2} - \frac{y_h^* + y_{*h}}{2}, \frac{z_{h+1}^* + z_{*h+1}}{2} - \frac{z_h^* + z_{*h}}{2} \right)$$

So now it is possible to define curvature and torsion as follows:

**Definition 2.** *For any* $h \in (1, \ldots, n-1)$*, the vector:*

$$\mathbf{K_h} = \mathbf{u_{h-1}} \times \mathbf{u_h} \tag{3}$$

*is called* ***vectorial curvature*** *of the h-th amino acid of P.*

**Definition 3.** *For any* $h \in (1, \ldots, n-1)$*, the vector:*

$$\mathbf{T_h} = \mathbf{K_h} \times \mathbf{K_{h+1}} \tag{4}$$

*is called* ***vectorial torsion*** *related to amino acid couple* $(s_P^h, s_P^{h+1})$ *of the h-th and (h+1)-th amino acids of P.*

This theory allows one to obtain the necessary information on the geometric structure of each consecutive amino acid quadruplet of a given protein *P*:

- Three distances $|C_i - C_{i+1}|$, $i \in \{h-1, h, h+1\}$;
- The angles $\varphi_h$, $\varphi_{h+1}$ determined by the vectors joining the centres;
- The angle $\theta_h$ corresponding to the two curvature vectors.

### 4.3. DBR$^2$ Scheme Definition

For the composition of an appropriate relational scheme, the use of a mixed strategy was chosen, combining the advantages of the top-down strategy with those of the bottom-up strategy. As a result of several phases of refinement, the Entity-Relationship (E-R) scheme realized is shown in Figure 3.

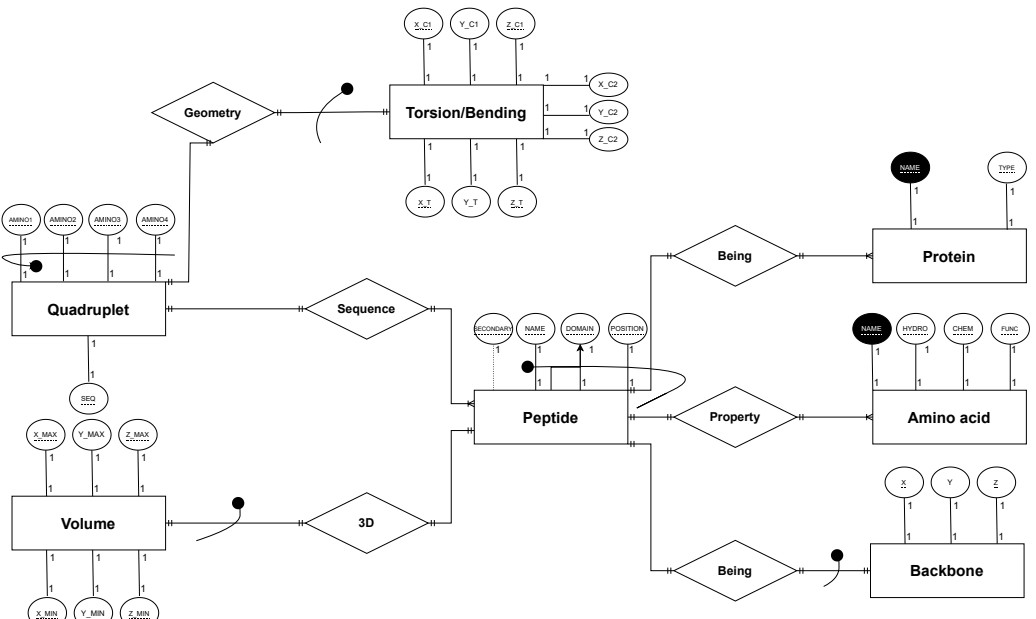

**Figure 3.** DBR2 ER scheme.

It shows a principal entity (`Peptide`) related to another six entities, four of these introducing geometric properties and mathematical models—these properties are fully described in [12] (`Volume`, `Quadruplet`, `Torsion/Bending` and `Backbone`)—and the other two describing amino acids' properties (`Protein` and `Amino acid`).

Moving from a conceptual to a logical schema is quite straightforward. The translation generates a schema composed of seven tables, as reported in Figure 4.

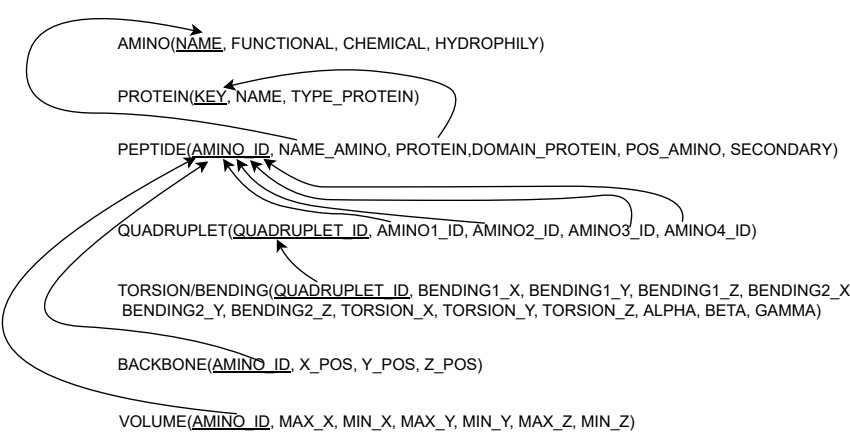

**Figure 4.** Conceptual scheme translation in logical scheme.

The last step consists of a physical implementation of the database called DBR$^2$ using PostGreSQL.

It is necessary to clarify an aspect of this implementation: as the logical scheme shows, the entity `Quadruplet` is linked to the entity `Peptide` using four foreign key constraints, but these are not implemented in the physical scheme. This choice is taken into account to avoid an error of referential integrity constraint violation due to a particular manipulation of control flows established while populating the database to simplify the operations. Indeed, the record related to the i-th amino of the chain is put in table `Peptide`, but during the same iteration of the 'for' cycle the record is also inserted that shows the i-th amino as the first of the quadruplet, and consequently the codes of the aminos (i+1)-th, (i+2)-th, (i+3)-th are required for this record, even if they have not already been inserted in the principle table `Peptide`.

*4.4. DBR$^2$ Instance Population*

The populating procedure has been very well articulated as shown in Figure 5. The number of the required steps is huge, as well as the processed items.

As already mentioned, population starts with the selection of the PDB files, available online at https://www.rcsb.org/ (accessed on 28 December 2022). As previously mentioned, the selection considers 446 Myoglobin proteins, and they are parsed one by one through the BioPython library. Thus, protein secondary structures are extracted from PDB files. Once all the information has been collected, they are inserted in the DBR$^2$. The information of interest is extracted to populate all the tables: `Protein`, `Peptide`, `Backbone`, `Volume` and `Quadruplet`. The remaining table `Torsion/Bending` is populated through several queries and calculations returning the geometrical parameter values. All these issues are addressed using the tool described earlier, following the schema described in Figure 5. This phase required a great effort and it is necessary to pursue the goal of the paper; since the data as initially presented in PDB show a lot of information that is not useful for our purposes and are not well interconnected to each other, it is not possible to easily manipulate them and represent the geometric model formulated. Therefore, it is necessary to use a software to simplify the manipulation of the data and the construction of the datasets for the analysis.

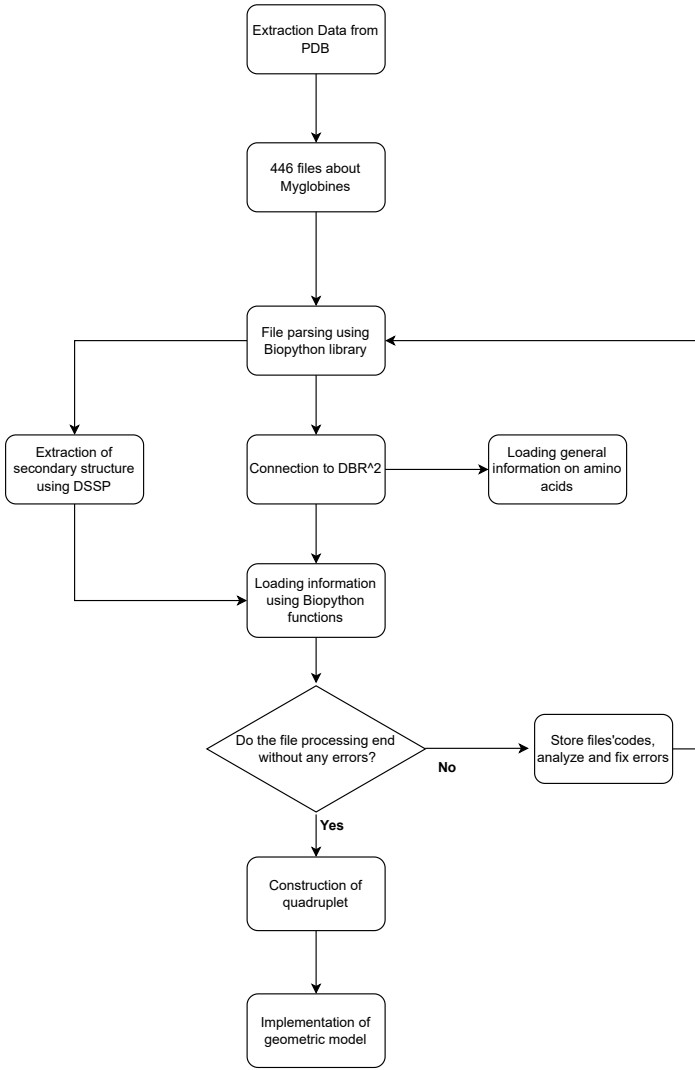

**Figure 5.** Illustrative schema of the DBR$^2$ population procedure.

## 5. Building the ML Models

### 5.1. Experiment Description

To conduct an analysis on DBR$^2$, it was necessary to build some additional tables, in which data were grouped and arranged in a way in which they could be more easily analyzed. It was decided to not insert such tables into the database schema but rather to store them in Comma Separated Value (CSV) files, processed through libraries offered by Python language, in subsequent scripts. The initial phase provides for the choice of features to be analyzed: in this case study, the target variable is the **amino acid volume**. The idea is to determine the different characteristics on which it depends. The fixed amino acid to be analyzed is named *Amino acid 0 (AA$_0$)*.

The features of interest are:

- Volume of the amino acid located in position $-3, -2, -1, 1, 2, 3$ from AA$_0$;
- Functional class of the amino acid located in position $-3, -2, -1, 1, 2, 3$ from AA$_0$;
- Chemical class of the amino acid located in position $-3, -2, -1, 1, 2, 3$ from AA$_0$;
- Hydrophilic class of the amino acid located in position $-3, -2, -1, 1, 2, 3$ from AA$_0$;
- Relative position of AA$_0$ in the chain;
- Norm of the 1st and 2nd curvature of the quadruplet in which AA$_0$ is in position 1;
- Norm of the 1st and 2nd curvature of the quadruplet in which AA$_0$ is in position 4;
- Norm of the torsion of the quadruplet in which AA$_0$ is in position 1;
- Norm of the torsion of the quadruplet in which AA$_0$ is in position 4;

- Volume of $AA_0$

At a later stage, the dataset is extended considering also the properties of the amino acids that reside in $AA_0$, comprising six and then nine residues that precede and follow it. To better clarify which are the features selected, how they have been processed and what is their geometrical meaning, it is possible to analyze the following schema Figure 6. Here is shown, as an example, what the steps are that allow one to populate the CSV file for the analyses. In this example, the amino acid Aspartic Acid (ASP) was considered in all the primary structures loaded in $DBR^2$. In particular, during STEP 1, the algorithm will detect its presence in the protein PDB code: 3HEN Listing 1, in position 4/153, 20/153 and so on. Then, in STEP 2, for each of these occurrences, the algorithm will also detect the two quadruplets—i.e., those in which ASP is in first and last position—and subsequently, during STEP 3, it will extract the geometrical features—according to the model presented in [12]—and the biological ones. Finally, in STEP 4, for every occurrence of ASP, a row, including all the information extracted, will be added to the CSV file.

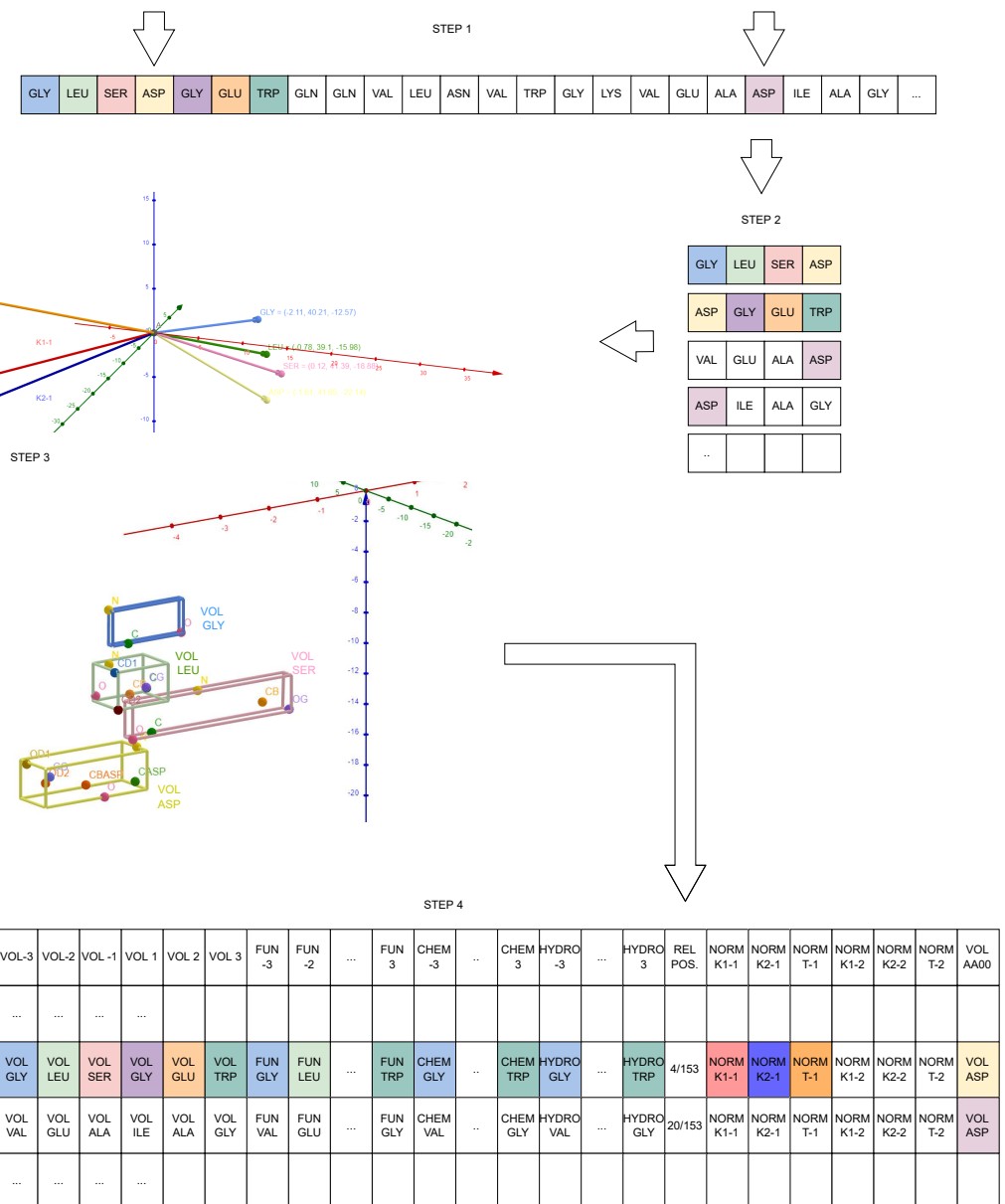

**Figure 6.** The schema of the dataset construction. The colors of the cells highlight the connection between the amino acids and the relative features with the same colors.

### 5.1.1. Lysine Analysis

Initially, attention was paid, for statistical and computational reasons, to the most frequent amino acid found in the myoglobins: lysine (LYS).

### 5.1.2. All Aminos, Separated

At the end of this phase, in order to make the analysis more complete, to generalize the results, obtained for LYS, to all the amino acids and to test the calculation capabilities of the server used, new CSV files are generated, one for each of the 20 amino acids. In all, repeating the same procedure described, 60 tables were created to conduct the analysis. In other words, for each amino acid—whatever it is, not only the Lysine—three files were analyzed, storing features of the residues in its surroundings, including the 3, 6 and 9 that precede and follow it.

### 5.1.3. All Aminos

Finally, the same approach was applied to create a unique big CSV file that stores the described information about all the amino acids of each myoglobin, without subdividing them. For reason of computational costs, this kind of analysis is limited to the features of three residues that precede and follow every $AA_0$.

### *5.2. Analyzing Data with RF*

To ensure a high network performance, a preliminary analysis was conducted on the parameters of the algorithm. An automatic model generally has several parameters that are not trained by the learning set and they control the accuracy of the model. Thanks to the `GridSearchCV()` function of the SKlearn library, it is possible to train the model on a grid built combining the different parameters with each other in all possible ways to find the best match of them, assuring the highest accuracy on target variable prediction.

This test was repeated for all the datasets described:

- Table that contains features about 3 amino acids preceding and following LYS;
- Table that contains features about 6 amino acids preceding and following LYS;
- Table that contains features about 9 amino acids preceding and following LYS;
- Table that contains features about 3 amino acids preceding and following each residue;

Of course, the kinds of parameters depend on the algorithm considered.

In the case of RF, it was decided to train the model by varying:

- The number of estimators: Random Forest trees may vary from 50 to 500, with step 10;
- The depth of the tree: the maximum number of children from root to node further leaf, can range from 5 to 12. The value None was also attached to the list, which means default maximum depth was not chosen.

For each parameter combination, the `evaluate()` function returns the model accuracy, so is possible to detect the best combination, Listing 2 presents such a situation.

**Listing 2.** RF hyperparameter optimization results.

```
LYS -3: {'max_depth': None , 'n_estimators': 300}
    Model Performance
    Average Error: 9.1795 degrees.
    Accuracy = 81.75%

LYS -6 : {'max_depth': None , 'n_estimators': 300}
    Model Performance
    Average Error: 8.6356 degrees.
    Accuracy = 82.04%

LYS -9: {'max_depth': None , 'n_estimators': 300}
    Model Performance
    Average Error: 8.5398 degrees.
    Accuracy = 83.39%
```

Since this process is long and expensive, the tuning of the hyperparameters was carried out only on the Lysine tables. It was chosen, however, to evaluate the model with the fixed hyperparameters obtained on Lysine, on all other tables. This process has reported excellent results in terms of accuracy, shown in Table 1:

**Table 1.** Table showing that for every amino acids the results, in terms of accuracy and average error, were performed training the RF with the hyperparameters found with the stress test.

| Amino Acids | Features about 3 Amino Acids Preceding and Following $AA_0$ | | Features about 6 Amino Acids Preceding and Following $AA_0$ | | Features about 9 Amino Acids Preceding and Following $AA_0$ | |
|---|---|---|---|---|---|---|
| | Average Error | Accuracy | Average Error | Accuracy | Average Error | Accuracy |
| GLY | 0.5385 | 75.54% | 0.5151 | 70.97% | 0.4468 | 78.68% |
| ALA | 0.7402 | 93.28 % | 0.6254 | 93.97 % | 0.6009 | 94.01% |
| VAL | 1.8636 | 92.94% | 1.4999 | 94.18% | 1.5162 | 93.98% |
| LEU | 2.8407 | 93.65% | 2.5713 | 94.23% | 2.4643 | 94.42% |
| ILE | 2.6804 | 93.02% | 2.4010 | 93.74% | 2.2002 | 94.25% |
| MET | 3.8676 | 92.29% | 2.8892 | 94.59% | 3.0405 | 94.29% |
| SER | 1.0704 | 93.76% | 1.0899 | 93.05% | 1.0885 | 93.25% |
| PRO | 1.6259 | 91.33% | 1.4459 | 92.46% | 1.3758 | 92.71% |
| THR | 1.6654 | 93.65% | 1.4738 | 94.31% | 1.4648 | 94.33% |
| CYS | 4.3423 | 71.34% | 3.4113 | 77.39% | 1.6509 | 93.66% |
| ASN | 2.4039 | 93.87% | 2.0666 | 94.81% | 1.8991 | 94.69% |
| GLN | 5.9150 | 88.48% | 4.4296 | 91.30% | 5.1124 | 91.34% |
| PHE | 4.6112 | 93.14% | 3.5674 | 94.76% | 3.4181 | 94.65% |
| TYR | 5.1439 | 94.54% | 4.0024 | 95.63% | 4.6987 | 95.48% |
| TRP | 4.8437 | 95.60% | 4.3107 | 95.98% | 4.2929 | 95.80% |
| LYS | 9.1833 | 81.72% | 8.5886 | 82.15% | 8.5366 | 83.39% |
| HIS | 3.3231 | 93.25% | 3.1346 | 93.60% | 2.8392 | 94.33% |
| ARG | 10.5045 | 69.72% | 10.1136 | 68.02% | 10.1957 | 68.46% |
| ASP | 2.8097 | 92.15% | 2.5156 | 93.04% | 2.5252 | 93.00% |
| GLU | 5.2578 | 88.68% | 5.1794 | 88.16% | 4.9968 | 89.58% |

In a first approach to the analysis, through the use of RF, it was chosen to predict the volume of each occurrence of 8821 Lysine stored in $DBR^2$, using features as its relative position within the chain, volumes and functional, chemical and hydrophilic properties of the 3, 6 or 9 previous and subsequent amino acids.

The created function, `random_forest(csvfilename)`, requires as input name the ray value of the $AA_0$ around it to be analyzed. Using the `train_test_split()` function allows splitting the dataset into a training set and a test set; in particular, having set the value of the parameter test size on 0.25, 70% of the set will be devoted to training and 30% to phase testing. After training the regressor on the training set, the `predict` method uses the predictor on the X_test to derive the y_test and, by means of the `evaluate()` function, the goodness of the model is estimated. In addition to the information on the goodness of the model, other information was searched for and saved: mean absolute error (MAE), mean squared error (MSE), root mean squared error (RMSE), the minimum between the max_depth of the RF trees, its number of nodes and arcs, the iterations made and the time taken to calculate such details. Listing 3 reports this case.

Because hyperparameter research led to the choice of training the regressor without imposing conditions on the maximum depth of forest decision trees, the graphical representation of the latter is very complicated: it is not presented in this paper, even if all the material can be sourced from the supplementary section of the paper and downloaded from the GitHub repository.

**Listing 3.** RF performance results.

```
LYS-3 {'max_depth': None, 'n_estimators': 300}:
    Average Error: 9.1833 degrees.
    Accuracy = 81.72%.
    Means absolute error   9.183323451548308
    Means squared error   186.71606785645264
    Root means squared error   13.664408800107402
    Minimum max depth of tree =   30
    16737 nodes 25068 edges 8420 iter 13.43~sec

LYS-6 {'max_depth': None, 'n_estimators': 300}:
    Average Error: 8.5886 degrees.
    Accuracy = 82.15%.
    Means absolute error   8.588595380071665
    Means squared error   166.42968189426642
    Root means squared error   12.900762841563534
    Minimum max depth of tree =   32
    16789 nodes 25137 edges 8155 iter 13.78~sec

LYS-9 {'max_depth': None, 'n_estimators': 300}:
    Average Error: 8.5366 degrees.
    Accuracy = 83.39%.
    Means absolute error   8.536609032254626
    Means squared error   165.80542920374762
    Root means squared error   12.876545701536093
    Minimum max depth of tree =   31
    14937 nodes 22372 edges 7057 iter 8.21 sec
```

### 5.3. Analyzing Data with MLP

The first step, mandatory to ensuring a high accuracy, is the hyperparameter research, as described in Section 5.2. In the case of MLP, it was decided to train the network by varying:

- The size of the hidden layers, from 100 to 1000 with step 300;
- The activation function, deleting from the grid those that did not carry, in any case, to the algorithm convergence;
- The solver, choosing the one compatible with the activation functions defined;
- The maximum number of iterations, from 100 to 1000 with step 200.

At the end of the process, the `best_estimator()` function returns the best combination detected, reported in Listing 4.

**Listing 4.** MLP hyperparameter optimization results.

```
LYS-3
    Parameters:
    {'activation': logistic, 'hidden_layer_sizes': 700,
    'max_iter':900, 'solver': 'adam'}
    Accuracy = 79.22%

LYS-6
    Parameters:
    {'activation': logistic, 'hidden_layer_sizes': 700,
    'max_iter':900, 'solver': 'adam'}
    Accuracy = 80.35%

LYS-9
    Parameters:
    {'activation': logistic, 'hidden_layer_sizes': 700,
    'max_iter':900, 'solver': 'adam'}
    Accuracy = 81.27%
```

This test required an expensive computational effort and an hour and half of computing for every file, but it was necessary to ensure the best fit to the algorithm. For this reason, the same approach used for RF was performed: it was chosen to apply the hyperparameters

found for the LYS to all the amino acids to avoid performing the stress test many times, even because the results of accuracy reached were reasonable, as expected. The results are shown in Table 2.

**Table 2.** MLP performance table.

| Amino Acids | Features about 3 Amino Acids Preceding and Following $AA_0$ | | Features about 6 Amino Acids Preceding and Following $AA_0$ | | Features about 9 Amino Acids Preceding and Following $AA_0$ | |
|---|---|---|---|---|---|---|
| | Average Error | Accuracy | Average Error | Accuracy | Average Error | Accuracy |
| GLY | 0.4813 | 74.08% | 1.5946 | 30.03% | 1.6121 | 10.36% |
| ALA | 0.6569 | 93.62% | 2.1218 | 78.64% | 1.8956 | 80.09% |
| VAL | 1.7685 | 93.14% | 4.8140 | 80.44% | 4.1212 | 83.52% |
| LEU | 3.2014 | 92.66% | 6.8586 | 85.08% | 6.7312 | 84.92% |
| ILE | 2.6705 | 92.96% | 6.8473 | 82.38% | 6.8182 | 82.39% |
| MET | 4.1665 | 92.59% | 7.8958 | 84.33% | 7.6145 | 86.10% |
| SER | 1.0998 | 93.36% | 2.7085 | 83.28% | 2.8237 | 82.77% |
| PRO | 1.5146 | 92.02% | 3.9080 | 79.57% | 3.7529 | 81.58% |
| THR | 1.7111 | 93.39% | 3.8131 | 85.29% | 3.7133 | 85.47% |
| CYS | 1.2494 | 95.18% | 3.4752 | 80.31% | 2.0410 | 91.85% |
| ASN | 2.1876 | 93.87% | 4.1435 | 89.31% | 3.9476 | 89.00% |
| GLN | 5.6628 | 90.25% | 11.1820 | 77.89% | 10.7624 | 81.91% |
| PHE | 3.8403 | 94.11% | 11.4726 | 83.08% | 12.1458 | 81.15% |
| TYR | 7.1454 | 93.21% | 12.9676 | 86.32% | 14.0738 | 85.72% |
| TRP | 5.5169 | 94.84% | 15.7977 | 85.43% | 8.6417 | 91.98% |
| LYS | 9.4384 | 81.81% | 16.5806 | 67.33% | 15.6154 | 70.29% |
| HIS | 3.3458 | 93.27% | 8.5671 | 83.00% | 7.5599 | 85.21% |
| ARG | 10.2835 | 71.13% | 17.4146 | 60.54% | 15.9363 | 61.93% |
| ASP | 2.7899 | 92.36% | 5.6960 | 84.28% | 5.1691 | 85.76% |
| GLU | 5.4416 | 88.77% | 10.4980 | 78.21% | 9.7335 | 79.30% |

## 6. Explainability Analysis and Discussion

### 6.1. Lysine Analysis

As mentioned in Section 5.1, the first step of the analysis is focused on training the two analysis algorithms on the files concerning the LYS. This script requires a parameter input, which can be '3', '6' or '9'.

In fact, based on this choice, one of the three files related to the LYS—containing the characteristics of the 3, 6 or 9 amino acids, respectively, previous and subsequent—are processed. The first instructions aim to select the features to be considered, based on the input indicated. In the case of using MLP, after training the network, to determine the feature importance, it was necessary to use the Python tool Eli-5 to rebuild the network behaviour as described in Section 2.3. The results of this stage are three graphs for both the methods, which show which characteristics most influence the prediction of the Lysine volume, assigning them a score of importance between 0 and 1.

The graph resulting from the analysis with RF shows a close dependence of the Lysine volume on the volumes of the surrounding amino acids. The same conclusions can be drawn even using the MLP algorithm. In addition, the increase in the number of surrounding amino acids considered confirms this pattern, at least in the case of Lysine, considering the results of analysis on files with the features of 6 and 9 residues around it.

However, there is a certain influence of the relative position feature in the prediction of the target variable. This sharp prevalence of volumes has caused some suspicions. A transitive dependence has been proposed: such volumes could depend on a third characteristic, which in this type of approach, however, does not emerge clearly and is put in the background.

Therefore, a speculative analysis was proposed, considering all the amino acids, in order to generalize the conclusions reached for the LYS and also to repeat the procedure, but eliminating all data concerning the volumes of the surrounding amino acids.

### 6.2. All Aminos, Separated

In order to dive deeply into the studies conducted on Lysine, it was immediately proposed to repeat the same procedure on each amino acid. The only difference involved in relation to the code allowing the analysis of the LYS is the introduction of an additional function that allows one to process one amino acid at a time rather simply by running a cycle. In the present case, it was decided not to use the graphic presentation of the results, because producing 20 different graphs does not make it easy to understand and compare them. It was decided to keep the scores of the six most important features for each amino acid and store them in a table containing the columns MAE, MSE and RMSE. These tables are present in the Supplementary Materials. From a careful analysis of these results, the volumes of the surrounding amino acids always emerge as the features that strongly influence the prediction of the target variable; however, it is possible to underline some different conclusions drawn using the two different methods. The results of the analysis carried out with RF lead to the following considerations:

- When the amino acid is considered in its configuration with three previous and subsequent residues, the variable that most affects the prediction is always one of the volumes of the surrounding residues, except in the case of aspartic acid in which the chemical property of a residue emerges previously. The following variables also apply for the greater part of the volumes.
- Even considering the information on the six residues prior to and after the $AA_0$, the volumes are the features that mostly influence the prediction. However, in the case of Aspartic Acid and Glutamine, their relative position in the chain acquires the role of main Features Importance (FI).
- Finally, considering a round of the $AA_0$ of 9 amino acid ray, the analysis reports, again, as first FI, one of the volumes of the surrounding amino acids, for almost all residues. In the case of aspartic acid and cysteine, in fact, the relative position is confirmed as the main FI and in the case of isoleucine the functional characteristic is of one of the following residues.

Despite that, something different emerges from the results of MLP analysis:

- When the amino acid is considered in its configuration with three previous and subsequent residues, the variable that most affects the prediction is always one of the volumes of the surrounding residues, with an exception made for cystein which shows the highest score in prediction for the geometric feature torsion of the first quadruplet. The following FIs are also, for the most part, volumes, but features related to the geometry still emerge.
- Even when considering the information on the six residues prior to and after the $AA_0$, the volumes are the most influential features of the prediction. This is the case even with the greater impact as compared to the previous case. However, in the case of proline and serine, torsion of a quadruplet in the chain acquires the role of the main FI.
- Finally, considering the surroundings of the $AA_0$ of the 9 amino acid ray, the analysis confirms the pattern established by the previous cases, reporting, again, as first in FI, one of the volumes of the surrounding amino acids, for almost all residues, without any exceptions. The geometrical features seem to appear as the third or following, in the classification of FI.

Considering, therefore, the hypothesis of a multiple dependence, it has been decided to repeat the analyses, eliminating all the data relating to the volumes of the surrounding residues—be they 3, 6 or 9—from the features on which to train the network. Even in that case, some analogies and some differences come out from the comparison of the two

approaches. The tables, obtained with RF and MLP analysis—presented in full in the Supplementary Materials—lead to the following considerations:

- On the one hand, the role of the relative position of the $AA_0$ stands out, as the main feature, showing its fundamental role in this type of analysis;
- On the other hand, contrary to what might be expected, the main FI that emerges when excluding volumes is not the characteristics of the amino acids (chemical, functional or hydrophobic), but the geometric ones, formulated ad hoc for the problem in examination. This has enhanced and supported the validity and correctness of the model created.

The results of MLP confirm the hypothesis, showing:

- Clear predominance of the geometric features related only to the torsion parameters, which decreases slightly only in the analysis of feature 9 residues in the surroundings.
- On the other hand, contrary to what might be expected—in complete disagreement with the same analysis conducted with RF—the other features that emerge when excluding volumes are not the other geometric characteristics—i.e., curvatures—but both relative position and the amino acid properties (chemical, functional or hydrophobic).

### 6.3. All Aminos

Considering only information about the three previous and subsequent amino acids from $AA_0$, the generated CSV file counted 68.613 rows and the analysis took several hours. Optimization of the hyperparameters, in the case of analysis with RF, produced the optimal accuracy in the case of `max_depth = None` and `n_estimators = 200`. With this setting, the `RandomForestRegressor()`, in a code completely analogous to the previous one, produced the following results:

```
TOT-3 {'max_depth': None, 'n_estimators': 200}:
   Average Error: 4.0070 degrees.
   Accuracy = 86.91%.
   Means absolute error:  4.010753088384853
   Means squared error:  62.97177931847423
   Root means squared error:  7.935475998229358
   Max_depth min =  42
   129749 nodes 194571 edges 70263 iter 1524.40 sec
```

As you can see, the accuracy is 86%, so you can consider the model very reliable. Also in this case, the FI are graphically represented. Concerning the analysis conducted with MLP, using the hyperparameters found with the stress test of Section 5.3, the degree of accuracy was lower but surely more than acceptable:

```
TOT-3: {'activation': 'logistic',
   'hidden_layer_sizes': 700, 'max_iter': 1000, 'solver': 'adam'}
   Average Error: 5.8446 degrees.
   Accuracy = 73.19%.
```

Both came to an unexpected and different conclusion from the previous one: the volume prediction of amino acid belonging to myoglobins, without distinction by type, is mostly influenced by its relative position within the domain. It reports a significantly higher score than that of the other features. It is clear that there is a close dependence of the volume of an amino acid on its relative position. All the characteristics concerning the volumes, indeed, have scores similar to each other, but barely more than half of the relative position. Despite what has been concluded for the relative position, this result is quite in disagreement with that reported in the analysis with RF, which shows a mix between volumes and amino acid chemical properties firmly among the first positions. As already happened at the end of the analysis carried out on the individual amino acids (see Section 6.2, second step comparison), curiosity has given rise to a final proposal for analysis. In complete analogy to the case study of the amino acids treated singularly, it was decided, in the end, to repeat the analysis, excluding the predominant feature: the relative position. Eliminating the main feature, will the volumes return to the top of the FI?

Using the optimal hyperparameters, the RF and the MLP, respectively, returned the following results:

```
TOT NO POS 3 {'max_depth': None , 'n_estimators': 200}:
    Average Error: 4.0616 degrees.
    Accuracy = 86.43%.
    Means absolute error:  4.07168620428757
    Means squared error:  64.37940508449833
    Root means squared error:  8.023677777958081
    Max_depth min =  42
    129765 nodes 194598 edges 69420 iter 1516.01 sec
```

```
TOT NO POS 3: {'activation': 'logistic',
    'hidden_layer_sizes': 700,'max_iter': 1000, 'solver': 'adam'}
    Average Error: 6.3096 degrees.
    Accuracy = 67.55%.
```

A key conclusion of FI was exactly the one expected. The results show that the main variables influencing volume prediction, using the RF model, excluding relative position, are still the volumes of the surrounding amino acids. Even the results obtained with MLP confirm the previous conclusions: there is a dependence of the volume of an amino acid from that of the surrounding residues. In this case, there is no clear distinction between the best feature score and that of the subsequent one. In addition, at volumes, the biological characteristics of amino acids and data related to torsion follow. This final result brought us to a twofold consideration:

- On one hand both, the models confirm the predominance of relative positions and volumes, so the main features seem to be well detached by the different approach.
- On the other hand, analysis with RF seems to underline the strength of the mathematical model; instead, analysis with MLP emphasizes the role of the torsion in determining the target variable and a quite significant impact of chemical properties.

## 7. Related Works

The scientific literature counts several works on eXplainable Artificial Intelligence (XAI): such works can be summarized in some comprehensive surveys, among which are [8,24].

While the comparison of ML approaches under the lens of performance and accuracy is a process that is quite often assessed in the scientific literature, there is a need for a unified and shared framework for measuring and comparing [25]. This need is motivated by the growing demand for a trustworthy AI [26]. Furthermore, the necessity of finding a good trade-off between accuracy and explainability is well known [27].

The present paper does not contribute to this theoretical discussion; rather, it presents a practical experience in determining the most meaningful features in a concrete, real-world case study; some of these papers are presented here.

- A comparative analysis of different Natural Language Processing (NLP) models in sentiment analysis of single domain Twitter messages [28]. In this paper, some ML models are analyzed and a comparison is made against classification accuracy and explainability capabilities.
- A study on the explainability in Deep Neural Network (DNN)s for image-based mammography analysis is reported in [29]. The paper makes a contribution with the introduction of an explainability method (named oriented, modified integrated gradients, OMIG) and its application to image analysis.
- The authors in [30] carry out a concrete analysis concerning the explainability of glaucoma prediction by merging the information coming from two different sources (tomography images and medical data).

The presented paper differs from the cited ones since it focuses on wide-ranging ML models and uses standard off-the-shelf technologies.

## 8. Conclusions and Future Developments

This paper presents a vertical experience, related to a biological problem, with a comparison between two mainstream ML models and explainability technologies.

The study confirms the presence of a trade-off between performance and explainability, as stated in [9]. RFs are highly explainable, while the accuracy of MLPs is better. The usage of the Eli-5 tool gives "non-natively explainable" formalisms—as MLPs—the same explainability power as other models. The usability of such a library is high.

As the discussion concerning the results of the experimentation reveals, there is a clear dependency of the predicted volume from the volumes of the surrounding amino acids and from its relative position in the chain, while there is not a meaningful dependency from the other geometrical features (e.g., torsion, bending, etc.). These features, of course, are expected to enter in case there are other problems to tackle, e.g., predicting the protein shape.

Another valuable result is that there are not, for the considered problems, meaningful differences between the two considered classifiers: RFs and MLPs. Differences are expected to arise in cases of more challenging problems and for another kind of model, i.e., DNNs.

This, however, constitutes a valuable result concerning the understanding of the minor differences of the FI mechanisms: one present in RFs and the other one coming from the usage of eli-5.

This notwithstanding, the results reported in this paper are limited to the scope of the paper itself, which is a study of the effects of XAI techniques. Such limitations can be summarized as follows:

1. The members of the chosen protein family—i.e., myoglobin—are very similar in their sequences (85–99% identity): this similarity implies that trained predictors exhibit high performance that is not met in reality; Otherwise, it was decided to perform this kind of analysis—aware of such similarity—because all the proteins belonging to the same family have similar functions. This similitude is intentional and allows one to study how geometrical features can vary within a similar primary structure. Moreover, it is clear that this has no impact on the results: the most important features are not directly related to the primary structure but to an amino acid's properties and its relative position in the chain (there could be multiple within the same structure).
2. The present work is based on the mathematical model reported in [10], which has its limited scope and constraining hypotheses: in particular, it does not explicitly consider gaps in the protein structure and does not use sophisticated bioinformatics methods for the evaluation of the protein structure.

Concerning such limitations, the validity of the results of the work is minorly impacted, not only due to the above-mentioned limitation of the scope of the work. In fact, the high-performance baseline for the prediction algorithm can exalt the differences between the two approaches (the first point); furthermore, the used mathematical model is fast and facilitates the computation of the population of the database.

Future research efforts will be devoted to using other explainability tools and libraries. A more formal framework to measure the explainability of trained models will also be considered in further analyses.

**Author Contributions:** Conceptualization, S.M.; methodology, S.M.; validation, E.B. and S.M.; investigation, data curation, software R.D.F. and R.D.G.; writing, R.D.F., R.d.G., E.B. and S.M. All authors have read and agreed to the published version of the manuscript.

**Funding:** This research received no external funding.

**Institutional Review Board Statement:** Not applicable because the work does not involve the use of animals or humans.

**Informed Consent Statement:** Not applicable because the work does not involve the use of humans.

**Data Availability Statement:** All the material is available on GitHub repository: https://github.com/robidfz/ProteinAnalysis.

**Acknowledgments:** The authors wants to thank the anonymous reviewers who spent a great effort in improving the paper with their continuous and competent remarks.

**Conflicts of Interest:** The authors declare no conflict of interest.

## Abbreviations/Code

The following abbreviations/codes are used in this manuscript:

| | |
|---|---|
| 3HEN | Ferric Horse Heart Myoglobin; H64V/V67R Mutant |
| $AA_0$ | Amino acid 0 |
| AI | Artificial Intelligence |
| CSV | Comma Separated Value |
| $DBR^2$ | Rosy and Roberta Database |
| DNN | Deep Neural Network |
| FI | Features Importance |
| LYS | Lysine |
| MAE | mean absolute error |
| ML | Machine Learning |
| MLP | Multilayer Perceptron Neural Network |
| MSE | mean squared error |
| NLP | Natural Language Processing |
| NN | Neural Network |
| PDB | Protein Data Bank |
| RF | Random Forest |
| RMSE | root mean squared error |
| V&V | Verification & Validation |
| XAI | eXplainable Artificial Intelligence |

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
