# Peer review of "Explainabilty Comparison between Random Forests and Neural Networks—Case Study of Amino Acid Volume Prediction"

_information, doi:10.3390/info14010021_

Round 1

Reviewer 1 Report

This is round 1 review of “Explainabilty Comparison between RFs and NNs: the Case Study of Amino Acid Volume Prediction” article by Stefano Marrone and colleagues, submitted in August 2022 for possible publication in “Information”.

The topic is interesting. It intersects ML research with bioinformatics from which it borrows input structures in order to compare results of prediction of conformational properties of residues from two groups (actually one) of monomers. It is commendable that Authors share their Python code and prepared the document in LaTeX. I did find, however, some major issues in this submission that must be addressed before it can be considered for publishing. They are listed below.

Main issues

I have a problem with the choice of “volume” of residues. Apparently it is calculated here as volume of smallest axis-aligned box delimiting the atoms (line 218). Despite being a pivotal factor, it is not properly defined and the paper that is the basis for the properties used (ref 4) does not mention box volume/size. Other approaches based on vdW radii and solvent exposure are not mentioned either. A box seems to be a loose approximation for volume, too. If I perform backbone rotation so that a sidechain is rotated 180 degrees from its current position, box volume won’t change much, but local conformation of the chain may be completely different. Even worse, everything changes if the whole molecule is rotated in any way. Moreover, I find it strange that common encoders for chain geometry (phi & psi dihedral angles) were not used as its descriptive features. These choices must be thoroughly explained by Authors.

The description of the database and methods is chaotic and makes it hard to follow the narrative. It should be clear (in this order) what is the goal, what is the data (source and initial processing), what are the tools (third party algorithms, etc), how those tools are linked in chains (workflow, etc), how it applies to the data, what are the results and how they tie to the goal and title. I would also like to see actual prediction on a sample structure. Effort should be focused on (but not limited to) Section 5.1.

Results find that "volume" of neighbor residues (sequence-wise) is the key factor to "volume" prediction of target residue. Myoglobins are alpha-only structures with nearly all residues located in helical regions which impose specific geometry on them, so it's not really surprising that they turn out to be locally "connected" (literally and figuratively). What happens if all volume features are dropped? Scientific value of this submission would increase if geometric properties could be - at least partially - predicted from non-geometric geometric features.

The Conclusions section is very short and poorly summarizes results of the presented research. In the Introduction Authors wrote that “While many scientific papers report theoretical evaluation of the different ML-based explainability techniques, there are few papers that tackle with real-world
problems.”, but there is no mention of the tackled real-world problem in Conclusions and how it ties with the advertised RF vs NN/MLP comparison. This section should be minimum thrice as long.

Portions of the text are dedicated to needless implementation details (e.g., do reader needs to know that Authors are using postgres?), but crucial information how many of the protein properties are calculated and processed is presented in a limited way. This is the first time this model is presented by Authors, so it is crucial to not withhold any details needed to reproduce the experiment. Increased page count can be offset by moving tables 3-14 to supplementary materials.

There are many problems with English that are detrimental to understanding of the text. Sometimes whole sentences make little sense. They are mentioned below, although only some. Text on flowcharts needs to be checked too (e.g., there is “Estraction” on Figure 5).

Other issues

[Title] For the benefit of wider spectrum of readers, consider changing the title to “Explainabilty Comparison between Random Forests and Neural Networks— Case Study of Amino Acid Volume Prediction”. It should still fit in two lines.

[Line 9] Elaborate about the input data set (i.e., how many and which PDB structures) and hint about reached conclusions. Half of current abstract is background and there is still some words left before the 200 word limit.

[Line 10] Should “neural network” and “machine learning” be added to the list of keywords?

[Line 12] There are no citations in the Introduction (there are only 13 references in the entire paper). They must be added here to place this work in the field and allow readers unfamiliar with the topic to find some good background resources. Try to integrate Section 7 here with all its citations plus maybe some more.

[Line 31] Define explainability here.

[Line 34] Rewrite “(…) the prediction of amino acid volumes in complex protein sequences” to simply “(…) the prediction of volume of protein residues.”. In the next sentence mention the data - chosen group of structures (myoglobins).

[Line 35] “(...) a subset of such database has chosen and preprocessed according to a method already present in the literature and then fed to the two different ML models” Apart from language, the method mentioned here is not cited.

[Line 41] Citation needed for (i.e., add some example papers) “While many scientific papers report theoretical evaluation of the different ML-based explainability techniques, there are few papers that tackle with real-world problems.”

[Line 60] There is actually ~195 000 structures in PDB at the time of this review.

[Line 69] It is customary that PDB structures mentioned / shown / analyzed directly in the main text are cited. Primary citation for 3HEN is <https://doi.org/10.1021/ja904726q> (this link can be found on its RCSB page). Also write “(3HEN)” in this line as “(PDB code: 3HEN)” or “(PDB ID: 3HEN)”. Later in the text it is fine to just use 3HEN.

[Line 70] The PDB format excerpt unnecessarily takes a whole page of the main text. Everybody who works with PDB files knows their structure. If Authors really (but really) want some of this file in the manuscript, trim it to HEADER, TITLE, COMPND, SOURCE, and ATOM for the first glycine residue (also decrease font size so there is no word wrapping in ATOM records).

[Line 89] All links can be moved from footnotes to bibliography as proper references (with time of online access).

[Line 127] No entry in ATOM record defines volume. What you have after XYZ coords is altloc occupancy and temperature factor <https://www.wwpdb.org/documentation/file-format-content/format33/sect9.html#ATOM>. These "other geometrical features" are actually statistical features.

[Line 131] Decision tree and random forest must be defined here. Also consider adding a figure with a sample tree here so that Figure 6 is more approachable. Maybe even put it side by side with a 3D structure of 3HEN as a “method and material” display (optional).

[Line 159] To make the paper shorter, Figure 1 (which shows a linear process) can be removed and steps shown in this subsection can be enumerated. Put differently, text only description is fine.

[Line 160] “Starting from the whole PDB, a first qualitative analysis is done to understand which is the subset of such a database that is thee subject of the study.” I barely understand this sentence. There are more language issues below, like “tasks of such a phase are”. From now on I’m no longer mentioning them in this review.

[Line 181] “Two distinct categories of proteins were chosen to be analysed: myoglobins and spikes.” What is a category in this sense? A search term in PDB, a domain from CATH/SCOP? It should be mentioned here how many structures were downloaded and sent to next stage (i.e., filtered). Their PDB IDs should be put in the supplement. What sequence identity threshold was used to eliminate duplicates? What was the X-ray resolution criterion?

[Line 194] Crystallographic experiments are imperfect and protein models in PDB often have gaps (missing residues), introducing local discontinuity in the chain. How Authors treated such situations? Also consider the fact that some residues can exhibit high mobility (expressed via b-factor ATOM property), which means their atom positions may vary from what is written in the file.

[Line 198] Authors copy terms from paper by Federica Vitale (ref 4) without explaining most of the symbols and refer the reader to that paper for more info. We are not given any samples of those values from a real or artificial structure. A figure displaying how these properties relate to a portion of a chain is welcome (with all important points and lines shown via fake atoms and bonds; PyMOL can render it easily).

[Line 215] Volume calculation is not defined.

[Line 234] In my opinion, focusing on myoglobins alone is enough for this paper. The other group has only 6 structures here and they are not used in the experiment. What’s the point of mentioning them then?

[Line 256] Some examples of values of the listed properties could be presented. For instance, what is the average “volume” of LYS in 3HEN or what is the curvature of a sample helix/sheet/loop

[Line 443] Instead of bunching separate but related figures together (e.g., 7-12), convert them to one figure with subfigures, with (a), (b), etc replacing “Features importance”. This will save some space for the mostly redundant captions.

[Line 450] “In the present case, it was decided not to use the graphic presentation of the results, because producing 20 different graphs does not make it easy for understanding and comparing them.” Supplementary materials will accept anything. Putting each data group in a separate file is a good idea.

[Line 465] All the colorful tables must be moved to supplementary materials, preferably to a dedicated file. This will considerably reduce size of the manuscript. One or two most important tables can be left in the main text.

[Line 636] 3HEN is not an abbreviation, but a PDB code (database accession number).

Author Response

I would like to thank the reviewer. The answers are in the attached letter.

Reviewer 2 Report

The paper presents explainable AI solutions built on random forest and genetic algorithm in multi layer perceptron. Rosy and Roberta Database is described.

Good presentation of the material.

However, I have a lot of questions:

1) why did Rosy and Roberta Database describe so detailedly? What is the reason for the paper's topic?

2) I think fig 6 has no value for the paper. It's hard to explain the result based on small letters.

3) Feature selection 7-16 is more important, but in this case, it would be great to demonstrate the predictive accuracy based on selected features and the whole dataset

4) Section Related works is unnecessary because it doesn't correlate with the rest of the paper.

5) Definition 4 is well-known

6) Statistical analysis of the dataset is more critical than fig 2-4

Author Response

(The authors gave the same response as above.)

Round 2

Reviewer 1 Report

While the manuscript has improved a bit, I am not entirely satisfied with its current state and some of the answers to my v1 report. Unimportant technical details are still in the spotlight (e.g., several depictions of db schemas), while fundamental aspects, such as proper demonstration of the model (i.e., transition from theory to practical application) and resolution of its issues with chain continuity and rotation of the molecule, have been hand-waived as “out of scope”, “not considered”, “not the main topic” or “not useful for evaluation of the results”. They are, in fact, very much in scope and must be considered. This stands in contradiction to how Authors “encourage the replication of the experiments from other researchers”. If the experimental setup is faulty, the results can be evaluated only as meaningless. Likewise, some of those results (structures, figures and tables) have now been relegated to Github instead of supplementary materials, which are the proper place for them that ensures their permanence.

Given the above, I cannot recommend acceptance of this submission. Not without a substantial revision that fully addresses all points listed below.

1. Authors are applying verbatim an old theoretical model (from 2007) to 400+ myoglobin structures, with “volume” of residues being a “predicted” property. When asked for the definition of a volume in this sense (i.e., the essence of this work), instead of explaining its calculation routine, They gave a reference to even older paper from 2005. Newer publications related to this topic are not cited, which suggests that those models did not gain traction in the field or are simply invalid. This is why I asked Authors to provide a complete description of the calculation procedure of the protein properties in order to show the readers how (and if) it works. They had to program all of this anyway, so there is little work to showcase it and refusing to do so raises a question whether the rest of data shown here is real. As a solution, it’s easiest to take 3HEN and make a couple of charts showing each parameter assigned to each residue (i.e., parameters from below line 245). Secondary structure and lysines should be marked. A companion 3D view is optional, but welcome. Any value can be assigned as atom’s temperature factor in Biopython and 3D viewers can color the protein in accordance with that.

2. The “volume” of a residue is assumed by the model to be a box containing its atoms. I asked how Authors dealt with the fact that a protein can be freely rotated, thus constantly changing the size of that box. They didn’t. If I rotate the entire database by any angle, the molecules will stay the same (i.e., the relative atom positions remain), but the residue "volume" calculated this way will be different. And structures in the PDB can be stored with any rotation, even if they represent exactly same molecule (sequence- and structure-wise). This must be normalized in some way or shown that it does not affect further processing.

3. Authors wrote that they didn’t consider gaps (i.e., missing residues) in the chain when calculating the radius of curvature or other properties. Take 1J3F from the db as an example. It has a 2-aa gap at K96 and H97 where there is no chain to work with (i.e., it is incorrect to just merge the ends of the gap). There maybe more such structures in the database - I haven’t checked. They should all be dropped or Authors should demonstrate that this does not affect the rest of the experiment.

4. Authors did not answer whether They have ensured that their protein database is non-redundant, e.g., by using a 30% sequence identity threshold. Hence, it is possible that there are duplicates. They must be removed for obvious reasons.

5. Authors insist on keeping those 6 spike proteins in the database “because they are part of the preprocessing phase” and “could be under study for future works”. It does not matter whether they are preprocessed or not because that does not affect the core of *this* work. They can be mentioned in conclusions as a future direction, but their inclusion in the main body is pointless and looks like false advertising.

6. Full list of PDB codes from the database must be provided in the supplementary materials to replicate the experiment. It can be a csv file, possibly with some additional details, like chain length, etc.

7. There is still not enough references (MDPI requires minimum 30 in a research paper). There are only two in the introduction and controversial statements - like in section 2.3 - are not backed up by citations.

8. Lastly, English of the manuscript did not improve much, contrary to the response. As an example, this sentence I mentioned last time is still here: [Line 40] “Starting from the well-known PDB, a subset of such database, including the group of myoglobin and spike proteins, has chosen and preprocessed according to a method already present in the literature [2] and then fed to the two different ML models”. Another example: [Line 123] “Since their born”.

Author Response

We thank the reviewer for the comments. The full letter containing answers are in attach.

Reviewer 2 Report

Dear authors, thank you for your answers. I have only one question about Related works section. In the present form, only descriptions of existing methods are given. From my point of view, it would be great to underline the drawbacks of the existing approach. In this case it will allow to underline the paper contribution

Author Response

The reviewer expressed a positive judgement on the paper. However, in attach the letter of comments regarding also the other reviewer.

Round 3

Author Response

I warmly thank the reviewer for all the effort spent on this paper. We added some consideration in the paper taking into account the comments. In particular, we clearly define the scope of the paper and synthesize the limitations of the approach in the conclusion section.

Reviewer 2 Report

I recommend the paper. Thank you

Author Response

I warmly thank the reviewer for all the effort spent on this paper.